# Bacteroidetes Species Are Correlated with Disease Activity in Ulcerative Colitis

**DOI:** 10.3390/jcm10081749

**Published:** 2021-04-17

**Authors:** Kei Nomura, Dai Ishikawa, Koki Okahara, Shoko Ito, Keiichi Haga, Masahito Takahashi, Atsushi Arakawa, Tomoyoshi Shibuya, Taro Osada, Kyoko Kuwahara-Arai, Teruo Kirikae, Akihito Nagahara

**Affiliations:** 1Department of Gastroenterology, Juntendo University School of Medicine, 2-1-1 Hongo, Bunkyo-ku, Tokyo 113-8421, Japan; ke-nomura@juntendo.ac.jp (K.N.); k-okahara@juntendo.ac.jp (K.O.); soyamada@juntendo.ac.jp (S.I.); khaga@juntendo.ac.jp (K.H.); matakaha@juntendo.ac.jp (M.T.); tomoyosi@juntendo.ac.jp (T.S.); otaro@juntendo.ac.jp (T.O.); nagahara@juntendo.ac.jp (A.N.); 2Department of Intestinal Microbiota Therapy, Juntendo University School of Medicine, 2-1-1 Hongo, Bunkyo-ku, Tokyo 113-8421, Japan; 3Department of Human Pathology, Juntendo University School of Medicine, 2-1-1 Hongo, Bunkyo-ku, Tokyo 113-8421, Japan; aatsushi@juntendo.ac.jp; 4Department of Microbiology, Juntendo University School of Medicine, 2-1-1 Hongo, Bunkyo-ku, Tokyo 113-8421, Japan; kkuwa@juntendo.ac.jp (K.K.-A.); t-kirikae@juntendo.ac.jp (T.K.)

**Keywords:** ulcerative colitis, microbiota, Bacteroidetes species, biomarker, Alistipes putredinis, Bacteroides stercoris, Bacteroides uniformis, Bacteroides rodentium, Parabacteroides merdae, Bacteroides thetaiotaomicron

## Abstract

Fecal microbiota transplantation following triple-antibiotic therapy (amoxicillin/fosfomycin/metronidazole) improves dysbiosis caused by reduced Bacteroidetes diversity in patients with ulcerative colitis (UC). We investigated the correlation between Bacteroidetes species abundance and UC activity. Fecal samples from 34 healthy controls and 52 patients with active UC (Lichtiger’s clinical activity index ≥5 or Mayo endoscopic subscore ≥1) were subjected to next-generation sequencing with *HSP60* as a target in bacterial metagenome analysis. A multiplex gene expression assay using colonoscopy-harvested mucosal tissues determined the involvement of Bacteroidetes species in the mucosal immune response. In patients with UC, six Bacteroides species exhibited significantly lower relative abundance, and twelve Bacteroidetes species were found significantly correlated with at least one metric of disease activity. The abundance of five Bacteroidetes species (*Alistipes putredinis*, *Bacteroides stercoris*, *Bacteroides uniformis*, *Bacteroides rodentium*, and *Parabacteroides merdae*) was correlated with three metrics, and their cumulative relative abundance was strongly correlated with the sum of Mayo endoscopic subscore (R = −0.71, *p* = 2 × 10^−9^). Five genes (*TARP*, *C10ORF54*, *ITGAE*, *TNFSF9*, and *LCN2*) associated with UC pathogenesis were expressed by the 12 key species. The loss of key species may exacerbate UC activity, serving as potential biomarkers.

## 1. Introduction

Inflammatory bowel disease (IBD), including ulcerative colitis (UC) and Crohn’s disease, represents a group of chronic inflammatory intestinal disorders resulting from complex interactions among genetic, immunological, and environmental factors whose etiology and pathogenesis are not fully understood [1,2]. In patients with UC, the diversity and richness of their intestinal microbiota are reduced, resulting in dysbiosis [3,4,5] and a significantly lower abundance of intestinal bacteria compared with that of healthy individuals [6,7]. Since changes in the microbiota can reflect disease activity, the abundance and diversity of intestinal microbiota may serve as a promising candidate biomarker for UC.

Fecal microbiota transplantation (FMT) is a therapeutic approach that is used to restore normal intestinal microbiota function by transplanting fecal bacterial microbiota derived from a healthy donor. FMT has been proposed as a form of microbial therapy for UC [8,9,10,11]. We previously reported that dysbiosis in the intestinal microbiota resulting from UC primarily results from a reduction in the number of Bacteroidetes operational taxonomic units and species diversity, resulting in the hyperproliferation and hypoproliferation of particular species. Moreover, we found that a single session of FMT following triple-antibiotic therapy (amoxicillin, fosfomycin, and metronidazole) reduced the symptoms of intestinal dysbiosis in patients with UC. This was achieved by the successful transplantation of live Bacteroidetes cells from donors, with both short-term efficacy and long-term maintenance of treatment [12,13,14]. Other reports have also suggested that improvement in the diversity and composition of Bacteroidetes species is beneficial for UC [15,16,17]. *Bacteroides thetaiotaomicron* can suppress inflammation in preclinical models of IBD [18]. Intestinal Bacteroides species have developed a commensal colonization system, which contributes to the homeostasis of gut microbiota [19], and reportedly synthesizes conjugated linoleic acid, which has immunomodulatory properties [20,21,22].

In this study, we aimed to determine the potential of Bacteroidetes species as a biomarker of UC. We compared composition of intestinal Bacteroidetes species between healthy controls and patients with UC, evaluated the correlation between Bacteroidetes species components and metrics of UC activity, and performed a multiplex gene expression assay to analyze the correlation between the intestinal mucosa of patients with UC and gene expression in Bacteroidetes species. We identified 12 key Bacteroidetes species that were significantly correlated with UC activity, as well as the expression of five genes involved in UC pathogenesis in colonic biopsy specimens. These findings can help identify specific microbial taxa and/or genes that can be used as reliable non-invasive fecal biomarkers, aiding the clinical management of UC.

## 2. Materials and Methods

### 2.1. Patients and Healthy Controls

Fifty-two patients with UC and 34 healthy controls were enrolled in this study from June 2014 to November 2017 at Juntendo University Hospital. UC was diagnosed based on standard clinical, endoscopic, and histological criteria. Eligible patients were over 16 years of age and had not received antibiotics or topical steroids, which can influence the microbiome composition, within 3 months before fecal sample collection [23,24]. Furthermore, all patients were confirmed to have active UC, as revealed by Lichtiger’s clinical activity index (CAI) ≥ 5 or Mayo endoscopic subscore (MES) ≥ 1. Patients with intestinal superinfections due to cytomegalovirus (determined by blood tests) were excluded from the study. Further exclusion criteria included pregnancy, current serious diseases, and participation in other clinical studies. The healthy controls were donor candidates from the clinical study of FMT. Health of donor candidates was ensured by the Juntendo donor screening criteria [12,13,14]. Donor candidates, who had been exposed to antibiotics within 3 months before the commencement of the study were excluded.

### 2.2. Fecal and Mucosal Sample Collection

Fecal and mucosal samples were collected on the same day. All fecal samples were transported to our laboratory within 6 h of collection, diluted 10-fold in TE buffer [10 mM Tris and 1 mM EDTA (pH 8.0)], and frozen at −80 °C until processing. Mucosal samples were taken from sections exhibiting the most severe inflammation by colonoscopy biopsy to be used for histopathology and gene expression assays.

### 2.3. Next-Generation Sequencing and Library Preparation

To identify key Bacteroidetes families, genera, and species associated with UC, we performed microbiome-wide analysis of partial sequences of heat shock protein 60 (*HSP60*), which was reported as a useful target sequence for bacterial metagenome analysis [25,26]. We have already reported the method for bacterial analysis involving DNA extraction, amplification by polymerase chain reaction (PCR), preparation of DNA libraries for next-generation sequencing, quality filtering of sequencing reads, and taxonomic analysis based on Bacteroidetes *HSP60* sequences [14]. In this study, we used a microbiome method targeting *HSP60*, utilizing the partial sequences widely employed in phylogenic analysis and species identification using Sanger sequencing, because of its higher diversity than that of 16S rRNA [27].

### 2.4. Multiplex Gene Expression Assay

Colonic mucosal tissues were obtained from inflammatory lesions of the colon from patients with UC through colonoscopy. Mucosal samples were pooled to generate 15 µL of each diluted sample (10 ng per primer pool). The multiplex gene expression assay (398 genes implicated in UC etiopathogenesis) was performed using the Oncomine Immune Response Research Assay (Thermo Fisher Scientific, Waltham, MA, USA) at the Nihon Gene Research Laboratories.

### 2.5. Statistical Analysis

All data were entered in Microsoft Excel and exported to GraphPad Software version 8.4.2. The mean and standard deviation values were computed for age, duration of disease, and clinical findings, whereas proportions were determined for sex and disease location. The composition of order Bacteroidetes were compared between healthy controls and UC patients. The statistical significance of differences upon pairwise comparisons was determined using the Mann–Whitney *U* test. We evaluated correlations between the relative abundance of Bacteroidetes species in fecal samples and UC activity through the following metrics: Sum of MES (in each part of the intestine: Periphery of the appendix vermiformis, cecum, ascending colon, transverse colon, descending colon, sigmoid colon, and rectum) [28], the Ulcerative Colitis Endoscopic Index of Severity (UCEIS), CAI, and Robarts histopathology index (RHI) [29]. However, we did not employ statistical correction for performing multiple comparisons that were not pre-specified as this increases the possibility of a false positive. Furthermore, we evaluated the correlation between the relative abundance of Bacteroidetes species and gene expression levels using Pearson’s correlation coefficient. Statistical significance was considered at *p*  ≤ 0.05.

## 3. Results

### 3.1. Patient Characteristics

The clinical characteristics of patients with UC are summarized in Table 1. On diagnosis, most patients presented with moderate to severe symptoms. From the *HSP60* PCR product DNA libraries, we obtained an average of 259,614, and 195,893 valid Bacteroidetes reads per sample from healthy controls, and UC patients. In addition, we obtained an average of 245,094, and 154,313 valid reads from the order Bacteroidales per sample, respectively. Bacteroidales is the predominant order in the phylum Bacteroidetes, and we previously reported that its species diversity is strongly correlated with the efficacy of FMT following triple-antibiotic therapy [14]. Therefore, we focused on the taxonomic composition belonging to the order Bacteroidales in the intestinal microbiota (Appendix A).

### 3.2. Differences in the Composition of the Order Bacteroidales between the Healthy Controls and UC Patients

Of the families, genera, and species belonging to the order Bacteroidales, the relative abundance of the following bacteria differed significantly between the two groups (Table 2). In terms of species, the relative abundance of 6 species (*Alistipes putredinis*, *Bacteroides coprocola*, *Bacteroides uniformis*, *Bacteroides cellulosilyticus*, *Bacteroides intestinalis*, and *Parabacteroides goldsteinii*) in healthy controls is significantly higher than that in UC patients.

### 3.3. Correlation between the Composition of Order Bacteroidales and the Metrics of UC Activity

The following 12 species (i.e., key species) were significantly associated with ≥1 metric of UC activity: Alistipes putredinis, Alistipes shahii, Bacteroides dorei, Bacteroides massiliensis, Bacteroides thetaiotaomicron, Bacteroides caccae, Bacteroides ovatus, Bacteroides stercoris, Bacteroides uniformis, Bacteroides rodentium, Parabacteroides merdae, and Parabacteroides distasonis (Table 3).

The abundance of *Bacteroides uniformis* (R = −0.60, *p* = 2 × 10^−6^) and that of seven other species was significantly correlated with the sum of MES. For *Bacteroides stercoris* (R = −0.54, *p* = 3 × 10^−5^) and six other species, their abundance was significantly correlated with UCEIS. Meanwhile, the abundance of *Bacteroides rodentium* (R = −0.44, *p* = 0.001) and six other species was significantly correlated with CAI while the abundance of *Alistipes shahii* and that of three other species was significantly correlated with RHI.

### 3.4. Cumulative Relative Abundance of Five Bacteroidetes Species Is Strongly Correlated with the Sum of MES

Among the 12 key Bacteroidetes species, the relative abundance of five species (*Alistipes putredinis*, *Bacteroides stercoris*, *Bacteroides uniformis*, *Bacteroides rodentium*, and *Parabacteroides merdae*) was correlated with three of four metrics of UC activity. The cumulative relative abundance of these five species was correlated with UCEIS (R = −0.58, *p* = 5 × 10^−6^) (Figure 1A), CAI (R = −0.28, *p* = 0.04) (Figure 1B), RHI (R = −0.36, *p* = 0.01) (Figure 1C), and strongly correlated with the sum of MES (R = −0.71, *p* = 2 × 10^−9^) (Figure 1D).

### 3.5. Correlations between Relative Abundance of the 12 Key Bacteroidetes species and Expression Levels of UC-Related Genes in the Colonic Mucosa

We evaluated the correlation between the expression of 398 genes involved in UC pathogenesis in colonic biopsy specimens and the relative abundance of 12 key Bacteroidetes species from our cohort. In total, the expression levels of 60 genes were correlated with the abundance of ≥2 Bacteroidetes species. A heatmap of these 60 genes was constructed, and the relative abundance of the 12 key Bacteroidetes species in the fecal microbiota was determined (Figure 2, Appendix A).

## 4. Discussion

To our knowledge, this study is the first to investigate the correlation between intestinal microbiota dysbiosis and several metrics of UC activity. In patients with UC, 6 Bacteroides species exhibited significantly lower relative abundance when compared to that in the healthy controls, and 12 key Bacteroidetes species identified in this study displayed negative correlations with metrics of UC activity. Therefore, the loss of these species is suggested to result from UC exacerbation, as Bacteroidetes species adhering to the mucosal surface may be unable to inhabit the niche of the extensively damaged mucosa without sufficient mucin production in highly severe UC [30]. Moreover, the study showed that the cumulative relative abundance of five species was correlated with three of four metrics of UC activity, but the correlation was strongest with the sum of MES. This result suggests that fecal microbiota may be interpreted differently depending on the extent of disease.

Regarding interactions between key Bacteroidetes species and the intestinal immune response, *ITGAE* is involved in not only intestinal damage but also systemic tissue damage associated with inflammatory diseases, including autoimmune diseases [31,32,33], and *C10ORF54* is a potent negative regulator of T cell function, being expressed on hematopoietic cells and leukocytes [34]. The expression levels of these genes associated with regulatory functions against excessive inflammation showed an increasing tendency alongside the increase in the proportions of key Bacteroidetes species. *TNFSF9* is a co-stimulatory molecule expressed on T cells and natural killer cells upon activation [35,36,37], *LCN2* expression is reportedly upregulated in IBD patients [38,39], and the regulatory dynamics of fecal *LCN2* have been harnessed as a sensitive biomarker of gut inflammation, both in animal models and in patients with IBD [40,41,42]. However, the proportions of key Bacteroidetes species showed a tendency to associate with downregulated genes that are related to the exacerbation of intestinal inflammation. The genus Bacteroides, belonging to the phylum Bacteroidetes, produces short chain fatty acids, which increase the number of colonic Treg cells by encouraging the migration of extraintestinal Treg cells [43,44]. Specific Bacteroidetes species control inflammation with zwitterionic capsular polysaccharides, which are bacterial products that modulate T cells, such as by inducing anti-inflammatory interleukin-10-secreting Treg cells [45,46]. We found *Bacteroides thetaiotaomicron*, one of the key Bacteroidetes species that correlated with the sum of MES and CAI. The finding supports the fact that *Bacteroides thetaiotaomicron* suppresses inflammation of IBD [18]. Taken together, the findings of this study suggest that abundant key Bacteroidetes species have therapeutic potential; further studies on the gnotobiotic species of Bacteroidetes are required to elucidate the regulatory functions of these 12 key species.

This study has some limitations, such as the limited number of patients and analysis of the microbial composition of only the order Bacteroidales. Analyzing other taxa would enable elucidation of the interactions among the Bacteroides species and the gene expression under the mutual influence of other intestinal bacterial groups. In addition, analyzing the mucosa-associated microbiome and the gene expression of mucosal membrane tissue from patients in remission could offer a better understanding of the Bacteroides species potentially associated with recurrence.

## 5. Conclusions

We identified 12 Bacteroidetes species whose relative abundance was negatively correlated with UC activity and can potentially serve as useful microbial biomarkers to evaluate the disease activity of UC and its exacerbation. The identification of microbiome components associated with disease activity may lay the foundation for the establishment of a set of microbiota biomarkers, offering a non-invasive and accurate method to monitor UC and establish appropriate personalized treatments.

## Figures and Tables

**Figure 1 jcm-10-01749-f001:**
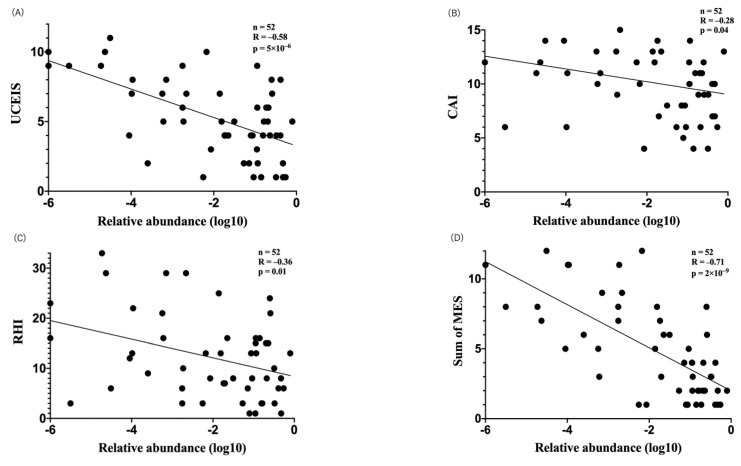
Correlation between the cumulative relative abundance of five key Bacteroidetes species and clinical evaluation. The cumulative relative abundance of five key Bacteroidetes species was correlated with UCEIS (R = −0.58, *p* = 5 × 10^−6^) (**A**), CAI (R = −0.28, *p* = 0.04) (**B**), RHI (R = −0.36, *p* = 0.01) (**C**), and the sum of MES (R = −0.71, *p* = 2 × 10^−9^) (**D**).

**Figure 2 jcm-10-01749-f002:**
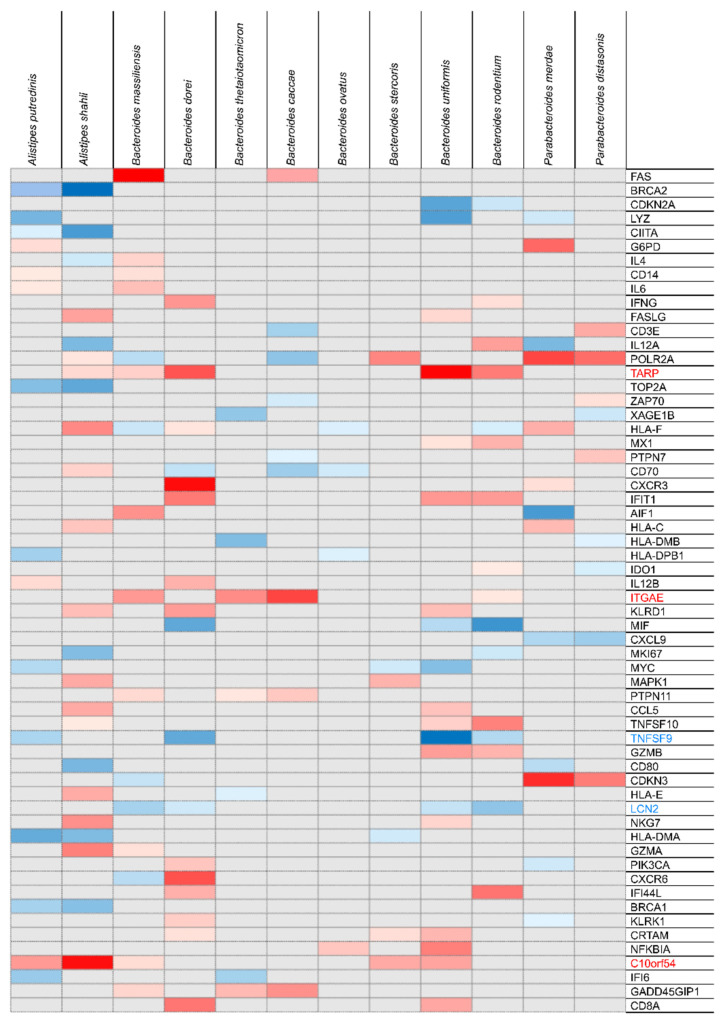
Heatmap of the correlation between 12 key Bacteroidetes species components and expression levels of ulcerative colitis-related genes in the colonic mucosa. The gray scale of the heatmap indicates no significant correlation at *p* > 0.05, while the red and blue scales indicate positive and negative correlations, respectively. The greater the correlation, the darker the color. *TARP* and *C10ORF54* (also known as *VISTA*) expression showed a significant positive correlation with five Bacteroidetes species, while that of integrin αE gene (*ITGAE* or *CD103*) was also significantly positively correlated with four Bacteroidetes species. Meanwhile, *TNFSF9* (also known as *CD137* or *4-1BB*) and lipocalin 2 (*LCN2*) expression was significantly negatively correlated with four Bacteroidetes species.

**Table 1 jcm-10-01749-t001:** Characteristics of patients with ulcerative colitis during sample collection.

Characteristic	Total (*n* = 52)
Age (years), mean ± SD	40.3 ± 13.4
Male/female, *n* (%)	35/17 (67.3)
Duration of disease (years), mean ± SD	9.0 ± 9.4
Disease location	
Proctitis, *n* (%)	10 (19.2)
Left sided colitis, *n* (%)	20 (38.5)
Extensive colitis, *n* (%)	22 (42.3)
Sum of MES, mean ± SD	5.0 ± 3.4
UCEIS, mean ± SD	5.3 ± 2.8
CAI, mean ± SD	10.2 ± 3.3
RHI, mean ± SD	12.2 ± 8.2

MES, Mayo endoscopic subscore; UCEIS, ulcerative colitis endoscopic index of severity; CAI, Lichtiger’s clinical activity index; RHI, Robarts histopathology index; SD, standard deviation.

**Table 2 jcm-10-01749-t002:** Differences in the composition of order Bacteroidales between the healthy controls and UC patients.

Taxonomic Description (Taxonomic Level)	Relative Abundance (%) (Standard Deviation)		Significance Level *t* Test
	Healthy	UC	
*Bacteroidaceae* (Family)	63.06 (28.311)	42.35 (32.969)	0.003
*Marinilabiliaceae* (Family)	0.05 (0.076)	4.56 (13.892)	0.024
*Bacteroides* (Genus)	62.56 (28.782)	39.90 (33.888)	0.002
*Alistipes putredinis* (Species)	2.10 (3.162)	0.05 (0.139)	<0.001
*Bacteroides coprocola* (Species)	5.73 (12.042)	1.24 (4.430)	0.047
*Bacteroides uniformis* (Species)	12.47 (11.960)	6.29 (9.143)	0.014
*Bacteroides cellulosilyticus* (Species)	0.26 (0.648)	0.002 (0.006)	0.026
*Bacteroides intestinalis* (Species)	0.09 (0.224)	0.0004 (0.002)	0.030
*Parabacteroides goldsteinii* (Species)	0.01 (0.028)	0.002 (0.009)	0.023

**Table 3 jcm-10-01749-t003:** Correlation between relative abundance of Bacteroidetes species in gut microbiota and metrics of ulcerative colitis activity.

Taxonomic Description (Taxonomic Level)	Relative Abundance (%) ± SD	Sum of MES (r, *p*)	UCEIS (r, *p*)	CAI (r, *p*)	RHI (r, *p*)
*Prevotellaceae* (Family)	13.40 ± 26.589	0.276, 0.048			
*Bacteroidaceae* (Family)	42.35 ± 32.969	−0.388, 0.004		−0.397, 0.004	
*Marinfilaceae* (Family)	0.69 ± 2.186	0.278, 0.046			
*Porphyromonadaceae* (Family)	28.63 ± 24.786			0.299, 0.031	
*Dysgonomonas* (Genus)	0.25 ± 1.202	0.287, 0.039			
*Bacteroides* (Genus)	39.90 ± 33.888	−0.433, 0.001		−0.459, <0.001	
*Macellibacteroides* (Genus)	0.24 ± 1.080		0.372, 0.007		
*Alistipes putredinis* (Species)	0.05 ± 0.139		−0.287, 0.039	−0.431, 0.001	−0.325, 0.021
*Bacteroides stercoris* (Species)	3.51 ± 9.550	−0.468, <0.001	−0.540, <0.001		−0.357, 0.011
*Bacteroides uniformis* (Species)	6.29 ± 9.143	−0.602, <0.001	−0.458, <0.001	−0.370, 0.007	
*Bacteroides rodentium* (Species)	0.31 ± 0.848	−0.394, 0.004	−0.313, 0.024	−0.440, 0.001	
*Parabacteroides merdae* (Species)	2.42 ± 5.691	−0.364, 0.008	−0.451, <0.001		−0.341, 0.015
*Parabacteroides distasonis* (Species)	16.68 ± 24.498	−0.307, 0.026	−0.347, 0.012		
*Alistipes shahii* (Species)	0.05 ± 0.139		−0.303, 0.029		−0.386, 0.006
*Bacteroides thetaiotaomicron* (Species)	1.41 ± 4.239	−0.344, 0.012		−0.383, 0.005	
*Bacteroides ovatus* (Species)	1.68 ± 5.004	−0.398, 0.003		−0.394, 0.004	
*Bacteroides caccae* (Species)	0.94 ± 3.551			−0.294, 0.034	
*Bacteroides massiliensis* (Species)	0.46 ± 2.292			−0.383, 0.005	
*Bacteroides dorei* (Species)	4.92 ± 11.895	−0.393, 0.003			

MES, Mayo endoscopic subscore; UCEIS, ulcerative colitis endoscopic index of severity; CAI, Lichtiger’s clinical activity index; RHI, Robarts histopathology index; SD, standard deviation.

## Data Availability

All data supporting this study are available in the article and its online Appendix A.

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
