# Peer review of "Bacteroidetes Species Are Correlated with Disease Activity in Ulcerative Colitis"

_jcm, 2021, doi:10.3390/jcm10081749_

Round 1
Reviewer 1 Report
In this paper, the authors attempt to determine the potential of Bacteroidetes species as biomarkers of UC disease activity by analysing samples from a cohort of 52 active UC patients. The authors identify some interesting correlations with Bacteroidetes
Comments
- the clinical design would benefit from control groups if attempting to determine the role of such species as biomarkers of disease activity ; suggest include group of healthy controls and also a group of UC patients in endoscopic remission / mucosal healing to allow more meaningful comparisons of how Bacteroidetes changes specifically correlate with UC disease activity
- the authors report on species level taxonomy classification despite use of amplicon sequencing rather than metagenomics
- can the authors comment on why they performed amplicon sequencing targeting hsp60 rather than 16S?
- despite performing numerous correlations without a pre-specified species / genus of interest, the authors did not correct for multiple comparisons
- do the authors have data on non-Bacteroidetes taxa that they can present to support that Bacteroidetes better correlates with disease activity and justifies specifically targeting them
Author Response
Reviewer1
In this paper, the authors attempt to determine the potential of Bacteroidetes species as biomarkers of UC disease activity by analysing samples from a cohort of 52 active UC patients. The authors identify some interesting correlations with Bacteroidetes
Comments
- the clinical design would benefit from control groups if attempting to determine the role of such species as biomarkers of disease activity ; suggest include group of healthy controls and also a group of UC patients in endoscopic remission / mucosal healing to allow more meaningful comparisons of how Bacteroidetes changes specifically correlate with UC disease activity
Response: We thank the reviewer for this constructive comment. In response, we have included the data with the comparison with 34 healthy controls as Table 2 “Differences in the composition of order Bacteroidales between the healthy controls and UC patients”. However, we regret to say that we do not have the data on a group of UC patients in endoscopic remission / mucosal healing.
The following sentence has been included in the Discussion as a limitation of this study:
“In addition, analyzing the mucosa-associated microbiome and the gene expression of mucosal membrane tissue from patients in remission could offer a better understanding of the Bacteroides species potentially associated with recurrence.”
- the authors report on species level taxonomy classification despite use of amplicon sequencing rather than metagenomics. Can the authors comment on why they performed amplicon sequencing targeting hsp60 rather than 16S?
Response: We thank the reviewer for this constructive comment. 16S rRNA-based microbiome analysis exhibits poor resolving power below the genus level. Therefore, a species-level analysis using a higher-resolution microbiome method is needed for identifying key bacterial species associated with diseases. In this study, we used a microbiome method targeting hsp60, utilizing the partial sequences widely used in phylogenic analysis and species identification using Sanger sequencing, because of its higher diversity, when compared to that of 16S rRNA. In response to this reviewer’s comment, we have included the reason for employing hsp60-based methods in the Materials and Methods section. The following sentence has been added:
“In this study, we used a microbiome method targeting HSP60, utilizing the partial sequences widely employed in phylogenic analysis and species identification using Sanger sequencing, because of its higher diversity than that of 16S rRNA.”
- despite performing numerous correlations without a pre-specified species / genus of interest, the authors did not correct for multiple comparisons
Response: We thank the reviewer for this constructive comment. We have included the data of genus and family level in Tables 2 and 3.
- do the authors have data on non-Bacteroidetes taxa that they can present to support that Bacteroidetes better correlates with disease activity and justifies specifically targeting them
Response: We thank the reviewer for this comment. We don’t have any microbial data on non-Bacteroidetes. In this study, we focused on Bacteroidetes species, because we previously reported that dysbiosis in the intestinal microbiota resulting from UC primarily results from a reduction in the number of Bacteroidetes OTU and diversity. We have added the following sentence in the Discussion as a limitation.
“This study has some limitations, such as the limited number of patients and analysis of the microbial composition of only the order Bacteroidales. Analyzing other taxa would enable elucidation of the interactions among the Bacteroides species and the gene expression under the mutual influence of other intestinal bacterial groups.”
Reviewer 2 Report
The manuscript by Nomura et al. presents data potentially meaningful for UC patients. Unfortunatly, the results obtained, possibly due to the limited patients number, are not convincing. The association between microbiota and gene pathway is only observational.
In vitro data to confirm the presented observation may increase the manuscript impact and significance.
Author Response
Reviewer2
The manuscript by Nomura et al. presents data potentially meaningful for UC patients. Unfortunatly, the results obtained, possibly due to the limited patients number, are not convincing. The association between microbiota and gene pathway is only observational. In vitro data to confirm the presented observation may increase the manuscript impact and significance.
Response: We thank the reviewer for this valuable comment. It is difficult to reach the conclusion that Bacteroidetes species serve as useful clinical biomarkers, because of the small number of cases analyzed in this study. To emphasize our findings, we have included the data from 34 healthy controls as a comparison in Table 2. We have also included the following sentence in the Discussion as a limitation. “This study has some limitations, such as the limited number of patients…”.
Unfortunately, we do not have any in vitro data. We have mentioned this in the Discussion. “further studies on the gnotobiotic species of Bacteroidetes are required to elucidate the regulatory functions of these 12 key species.”
Reviewer 3 Report
In the GI tract dominate microbiota are anaerobic bacteria. Very important role have Bacteroidetes, which are most numerous. During ulcerative colitis, especially in exacerbations, are changes in microbiota, however we still do not know the details, the role of individual bacteria and their relationships. This review is therefore very significant in getting to know these dependencies. Authors showed that abundance of five Bacteroidetes species was correlated with three metrics of ulcerative colitis.
I have some points for corrections:
- Authors obtained mucosal samples from sections exhibiting the most severe inflammation. However, interesting would be possible differences in abundance and the ratio of bacteria between severe inflammed and weakly inflammed tissues.
- In the article is lack of control group (!) of persons without colitis ulcerosa.
Author Response
Reviewer3
In the GI tract dominate microbiota are anaerobic bacteria. Very important role have Bacteroidetes, which are most numerous. During ulcerative colitis, especially in exacerbations, are changes in microbiota, however we still do not know the details, the role of individual bacteria and their relationships. This review is therefore very significant in getting to know these dependencies. Authors showed that abundance of five Bacteroidetes species was correlated with three metrics of ulcerative colitis.
I have some points for corrections:
- Authors obtained mucosal samples from sections exhibiting the most severe inflammation. However, interesting would be possible differences in abundance and the ratio of bacteria between severe inflammed and weakly inflammed tissues.
Response: We thank the reviewer for this valuable comment. We are also interested in the relationship between the abundance and the ratio of bacteria in the microbiota-associated mucosa and the non-inflamed mucosa from the patients in remission. We have included the following sentences in the Discussion. “In addition, analyzing the mucosa-associated microbiome and the gene expression of mucosal membrane tissue from patients in remission could offer a better understanding of the Bacteroides species potentially associated with recurrence.”
2. In the article is lack of control group (!) of persons without colitis ulcerosa.
Response: We thank the reviewer for this constructive comment. We have included the data with the comparison with 34 healthy controls as Table 2 “Differences in the composition of the order Bacteroidales between the healthy controls and UC patients”.
Round 2
Reviewer 1 Report
The authors have addressed most of my comments or acknowledged limitations. However some points remain to be addressed
1: Authors should provide reference(s) supporting their claim that "targeting HSP60, utilizing the partial sequences widely employed in phylogenic analysis and species identification using Sanger sequencing, because of its higher diversity than that of 16S rRNA"
2: The authors have still not employed appropriate statistical correction for performing multiple comparisons that were not pre-specified as this increases the possibility of a false positive; they should either do so or acknowledge in the text / table that they have not
Author Response
Reviewer 1
The authors have addressed most of my comments or acknowledged limitations. However some points remain to be addressed
1: Authors should provide reference(s) supporting their claim that "targeting HSP60, utilizing the partial sequences widely employed in phylogenic analysis and species identification using Sanger sequencing, because of its higher diversity than that of 16S rRNA"
Response: We thank the reviewer for this comment. In response, we have provided reference #27 supporting a microbiome method targeting hsp60 used in this study.“27. Sakamoto, M.; Ohkuma M. Usefulness of the hsp60 gene for the identification and classification of Gram-negative anaerobic rods. Journal of Medical Microbiology. 2010, 59, 1293-1302.”
2: The authors have still not employed appropriate statistical correction for performing multiple comparisons that were not pre-specified as this increases the possibility of a false positive; they should either do so or acknowledge in the text / table that they have not
Response: We thank the reviewer for this constructive comment. We are sorry that we did not understand the precise statistical comment from the reviewer 1 in first round. We have included the following a sentence in the Method. “However , we did not employ statistical correction for performing multiple comparisons that were not pre-specified as this increases the possibility of a false positive.”
Reviewer 2 Report
The revised version is significantly improved.
In vitro data would have been important, but I recognize that it is basically a new project requiring several months of experiments.
Author Response
The revised version is significantly improved. In vitro data would have been important, but I recognize that it is basically a new project requiring several months of experiments.
We thank the reviewer for understanding our situation. We have started basic experiments using gnotobiotic animal to evaluate the function of key Bacteroidetes species, however, we need considerable amounts of time and labor to show in-vitro data.
Reviewer 3 Report
Authors corrected manuscript substantially. Recently, I recommend it for publication.
Author Response
Authors corrected manuscript substantially. Recently, I recommend it for publication.
Thank you very much for providing important comments. We are thankful for the time and energy you expended. We have added a reference and sentences in the section of Method in revised version.
This manuscript is a resubmission of an earlier submission. The following is a list of the peer review reports and author responses from that submission.